# LasUIE: Unifying Information Extraction with Latent Adaptive Structure-aware Generative Language Model

**Hao Fei** [1]    **Shengqiong Wu** [1]    **Jingye Li** [2]    **Bobo Li** [2]    **Fei Li** [2]
**Libo Qin** [1]    **Meishan Zhang** [3*]    **Min Zhang** [3]    **Tat-Seng Chua** [1]

[1] Sea-NExT Joint Lab, School of Computing, National University of Singapore
[2] Wuhan University        [3] Harbin Institute of Technology (Shenzhen)
{haofei37, liboqin, dcscts}@nus.edu.sg    swu@u.nus.edu
{theodorelee, boboli, lifei_csnlp}@whu.edu.cn
mason.zms@gmail.com    zhangmin2021@hit.edu.cn

## Abstract

Universally modeling all typical information extraction tasks (UIE) with one generative language model (GLM) has revealed great potential by the latest study, where various IE predictions are unified into a linearized hierarchical expression under a GLM. Syntactic structure information, a type of effective feature which has been extensively utilized in IE community, should also be beneficial to UIE. In this work, we propose a novel structure-aware GLM, fully unleashing the power of syntactic knowledge for UIE. A heterogeneous structure inductor is explored to unsupervisedly induce rich heterogeneous structural representations by post-training an existing GLM. In particular, a structural broadcaster is devised to compact various latent trees into explicit high-order forests, helping to guide a better generation during decoding. We finally introduce a task-oriented structure fine-tuning mechanism, further adjusting the learned structures to most coincide with the end-task's need. Over 12 IE benchmarks across 7 tasks our system shows significant improvements over the baseline UIE system. Further in-depth analyses show that our GLM learns rich task-adaptive structural bias that greatly resolves the UIE crux, the *long-range dependence issue* and *boundary identifying*.

## 1  Introduction

Information extraction (IE) is widely considered as one of the most kernel topics in natural language processing (NLP), which is defined as to identify the desired structural information from the unstructured texts [4, 63, 47, 44, 39, 29, 15]. There is a variety of IE and IE-derived tasks, yet all of which revolves around predicting two key elements: *mention spans* or/and *their semantic relations*. For example as in Fig. 1(b), NER detects the mention spans, while RE recognizes each possible mention and its associated mention with relation. In this regard, all the existing IE jobs can be reduced into three prototypes: span extraction, pair extraction and hyper-pair extraction, as depicted in Fig. 1(a).

In the era of deep learning, IE witnesses extraordinary developments, where especially the recent triumph of pre-trained language models (LMs) helps push the state-of-the-art (SoTA) IE performances amazingly [10, 3, 25, 80, 73]. Prior related works mostly design particular models for certain IE tasks in isolation; while the latest SoTA progress [42] is achieved by unifying all IE tasks with a single encoder-decoder GLM, i.e., UIE. As different IE tasks essentially share the similar nature (i.e., modeling span and relation features), it is proven that universally modeling multiple IE tasks helps further learning of general sharable knowledge from varying task sources, which makes UIE great potentials in real-world scenarios. In this work we inherit this wisdom and also focus on UIE.

---

*Corresponding Author: Meishan Zhang

36th Conference on Neural Information Processing Systems (NeurIPS 2022).

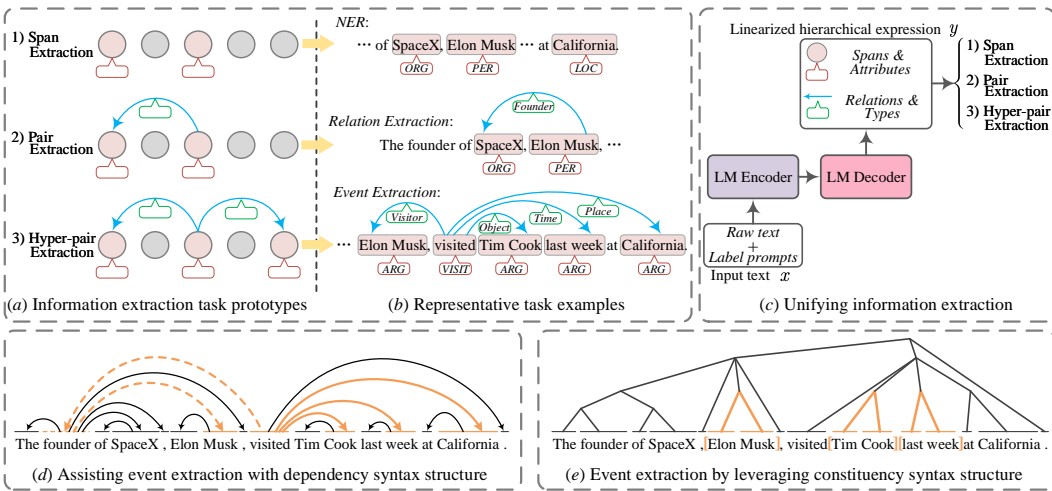

Figure 1: We reduce all the IE tasks into three prototypes (a) with representative examples (b). We unify all IEs with an encoder-decoder GLM (c). Both syntactic *dependency* (d) and *constituency* structure (e) plays a key but distinct role in IE, where the former helps solve *long-range dependence* problem and the latter benefits *boundary detection* issue. Best viewed with zooming in.

On the other hand, previous IE research extensively employs the external syntactic structure information, such as the dependency tree, for task improvements [5, 49, 45, 24, 52, 18]. Behind the enhancements is that IE structure corresponds much with the syntax structure explicitly, where the latter can essentially offer low-level linguistic bias for better learning the high-level semantic structure. As exemplified in Fig. 1(d), the dependency tree coincides much with structure of EE task as in Fig. 1(b). Importantly, some findings reveal that the LMs, being pre-trained on large corpus, capture structural syntax knowledge [68, 20, 23], which gives rise to LMs' distinguishing promotion on IE. Yet probing tasks show that the auto-learned structure representations is weak, which inevitably limits the LM efficacy for IE [8, 30, 65]. Correspondingly, a line of researches fuse external syntax trees into LMs to reinforce the structure awareness, i.e., structure-aware LMs [71, 2, 38, 6].

**Motivations.** After carefully revisiting the existing literatures, we summarize four key limitations of syntactic structure-aware LMs that hamper IE from further improvements. **First**, existing structure-aware LMs are mostly designed for one certain IE task (e.g., NER [69], RE [36]) instead of UIE, leaving the shared IE knowledge and the task-invariant syntax features unexploited. **Second**, current structure-aware LMs merely consider making use of one standalone type of syntax structures, i.e., mostly using the dependency trees [49, 24]. We however argue that as one core grammar, constituency syntax can serve complementary contributions for IEs. There are two common challenges of IEs: *long-range dependence problem* and *boundary identifying*, in which the dependency structure especially helps solve the former one [49, 45, 52] and the constituency syntax could mostly benefits the latter [79, 46, 54], as in Fig. 1(d)&(e). Thus it is best to simultaneously model both two heterogeneous structures [31, 17]. **Third**, existing works mostly integrate supervised syntax parse trees, where unfortunately, either the amount of manually annotated syntactic data (e.g., PTB) are largely limited, or the annotation noises from third-party parsers are inevitably introduced due to such explicit injection. **Fourth**, parsing syntax comes with task-irrelevant or indirect substructures (e.g., in Fig. 1(d) the black and dotted lines respectively), which would deteriorate the efficacy. Meanwhile, different IE tasks largely demand distinct bias of structural features, while current structure-aware LMs fail to fine-tune the structure knowledge to allow the structure bias best accord with end task's need.

**Contributions.** On the above basis, we propose learning a latent adaptive structure-aware generative language model for UIE (namely LasUIE). First of all, we reduce UIE into three uniform prototypes, upon which we transform the UIE into generative paradigm with an encoder-decoder GLM, predicting the linearized hierarchical expression (i.e., spans&attributes, relations&types, as shown in Fig. 1(c)). Then, we adopt a three-stage of LM training procedure, where an additional structure-aware post-training is added between the pre-training and fine-tuning stages for structure learning. Inspired by the progress of unsupervised grammar induction [58, 59, 28, 60], we design a heterogeneous structure inductor (HSI) module, where two heterogeneous syntactic structures are simultaneously measured

and automatically learned. With HSI, our GLM initialized with existing pre-trained parameters, during post-training, performs unsupervised syntax induction based on unlabeled texts without relying on external syntax parses or any annotation labor (cf. Fig. 2).

Since the induced latent structural representations may be squeezed aside by the mainstay contextual representations in LM encoder, we further enhance the utility of syntax by introducing a structural broadcaster (SB) module (cf. Fig.2). SB compacts multiple varying latent trees from different encoding attention heads into an explicit constituency-like and a dependency-like forest respectively. During each decoding step, two heterogeneous syntactic forests are utilized to produce high-order features at global level for guiding better content generation. Finally, during the prompt-based fine-tuning stage we perform task-oriented structure adaptive tuning to narrow the gaps between the induced syntactic and task-specific structures (cf. Fig. 3). With policy gradient we dynamically adjust the attributes of two heterogeneous structures according to the feedback of end task performance.

Extensive experiments are performed on 12 representative data across 7 IE tasks. On both the supervised and low-resource settings our framework consistently shows improvements over the baseline systems. Via further analyses we verify that **1)** unifying IE tasks by further modeling structure information in LM benefits IE substantially, especially in the low-resource scenario. **2)** Integrating two heterogeneous structures brings mutual advantages for UIE, helping fully resolve the boundary identifying and long-range dependence issue. **3)** Automatically inducing latent structures in LM with further task-oriented structural adaptation learning significantly consolidates the efficacy of structure knowledge for end tasks. **4)** Different types of IE tasks rely subtly on varying structural bias, all of which can be flexibly learned and correctly satisfied by our system. Our resources can be found at `https://github.com/ChocoWu/LasUIE`.

## 2 Related Work

IE is a long-standing research topic in NLP, which includes various tasks as well as growing derivations [4, 63, 47, 44, 35, 78]. We reveal that essentially all the IE tasks can be summarized into three main prototypes, according to the combination numbers of 'mention span' and 'semantic relation' prediction targets: 1) **span extraction**, e.g., named entity recognition (NER) [9], aspect-based sentiment analysis (ABSA) [64], aspect-term extraction (ATE) [37]; 2) **pair extraction**, e.g., relation extraction (RE) [81, 34], aspect-opinion pair extraction (AOP) [85], aspect-based sentiment triplet extraction (ASTE) [50]; and 3) **hyper-pair extraction**, e.g., event extraction (EE) [21], semantic role labeling (SRL) [19], opinion role labeling (ORL) [27, 61]. Mostly prior IE researches all solve one particular task exclusively (or one specific IE type) [49, 72, 40, 77, 86], while they may unfortunately ignore certain task-invariant universal IE features. In this work, we consider the line of UIE, unifying all IE tasks to exploit the shared IE knowledge. And based on the above UIE prototypes, we develop a LM-based unified framework with generative paradigm.

Many efforts are paid for building LMs to handle IE tasks by taking advantages of the knowledge from large-scale pre-training [10, 3, 25, 80, 73]. Another line of IE researches propose injecting external knowledge into LMs or GLMs, such as knowledge graph (KG) [41, 26, 83, 16], syntax structure information [71, 2, 38, 6]. Comparing to the integration of domain-specific KG information for certain IE tasks, syntactic information would provide much broader generic features in the scope of UIE. The very latest research attention of LMs has been focused on the GLMs, the encoder-decoder paradigm LMs. GLMs transform various NLP tasks into a unified seq-to-seq scheme with some properly-designed prompt texts as additional inputs [32, 55, 80]. Very recently, Lu et al. (2022) [42] pioneer the UIE by casting the IE structure prediction into text generation with a GLM, with which our UIE modeling shares the same spirit. We however note that our work can advance in two major aspects. First, we consider the integration of additional structural knowledge in GLMs for UIE enhancements. Besides, [42] require supervisedly pre-training their UIE GLM on a large-scale annotated IE corpus, while our system automatically induces structure knowledge based merely on unlabeled texts without any further annotation and labor.

This work also closely relates to the line of structure-aware LMs. On the one hand, some researches propose directly introducing external syntax trees into LMs to reinforce the structure awareness. They mostly take the Transformer-based LMs as backbone, and fuse the syntax signals (annotations) from external parsers or PTB corpus by modifying the Transformer attentions [71, 38, 6]. Another line of structure-aware LMs directly induce syntax structure into LMs automatically, a.k.a., unsupervised grammar induction [7, 75, 58, 12, 59, 28]. We in this work borrow the success of unsupervised

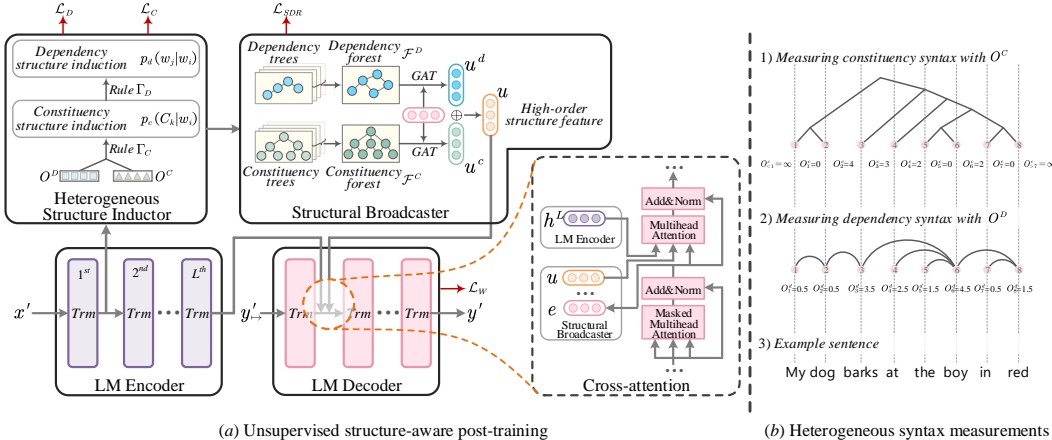

Figure 2: Our LasUIE framework under (a) unsupervised structure-aware post-training ($y'_{\mapsto}$ refers to prediction $y'$ with shift right in seq-to-seq procedure). Heterogeneous structure inductor module generates both constituency and dependency structures via (b) two heterogeneous syntax measurements.

grammar induction, inducing rich structure information for LMs for better UIE. Inspired by the foundation of syntax distance measurements [58, 13, 60], we propose to induce linguistic structures and compose both the constituency and dependency syntax structures simultaneously.

## 3 Unifying Information Extraction with Text Generative Paradigm

As aforementioned, we reduce all IE jobs into the predictions of few structural elements: 1) *spans* and 2) *relations*, and without losing the generality of UIE, we also consider the predictions of 3) *span attributes* and 4) *relation types*. Using different combinations of the structural elements properly can construct a hierarchical IE structure. In other words, we can arrange these elements into a sequential textual expression, from which the actual IE target can be easily restored. Based on this we unifiedly model all IE tasks (i.e., UIE) by transforming the structure prediction into text generation. Fig. 1(c) illustrates the main idea. Such generative scheme also enables to take the advantage of the recent achievement of GLMs, such as BART [32] and T5 [55]. Taking an existing GLM as backbone, we reach the goal of end-to-end UIE as well as complex IE, such as overlapped and discontinuous cases [82, 14, 33]. Our LM encoder takes input text (i.e., $x$), and the decoder produces the linearized hierarchical expression (LHE), i.e., $y$. Fig. 3 illustrates the input and output implications.

**Input.** The input text $x = \{w_1, \cdots, w_n\}$ includes the raw sentence and task-specific label prompts. The label prompts contain the pre-defined task-specific labels, including *span attributes* ('*Attr*') and *relational types* ('*Type*'), where each label is separated by a '' or a '<REL>' marker. Parts of the '*Attr*' and '*Type*' labels will be copied and output in $y$. We also insert a task identifier token '<TASK>' to inform the model which task to predict.

**Output.** The output text $y$ is a linearized hierarchical expression that describes how the structural elements organize into the target structure, as depicted in Fig. 3. For example, in span extraction, $y$ should be a list of text spans and attribute labels, i.e., '$\{(Span, Attr), \cdots\}$'. In pair extraction $y$ is a list of pairs, where a pair is represented as '$(Span_i, Attr_i [Type_k](Span_j, Attr_j))$' in which $Span_j$ is a subordinate mention of $Span_i$ with a semantic relation $Type_k$. For hyper-pair extraction $y$ is a list of hyper-pairs represented as '$(Span_i, Attr_i [Type_k](Span_j, Attr_j) [Type_m](Span_k, Attr_k) \cdots)$'.

It is also noteworthy that our LHE takes a similar scheme with [42], but with difference. For example, in our scheme all the mention comes with an associated attribute label in any IE prototype; while in [42] the subordinate mentions have no attribute labels. Thus, our design could be more generalized.

## 4 Learning Latent Adaptive Structure-aware Generative Language Model

### 4.1 Overall Framework

The overall framework is built upon a Transformer-based encoder-decoder GLM, based on which we additionally add 1) a heterogeneous structure inductor module at top of the encoder for structural

learning, 2) a structural broadcaster module between GLM encoder and decoder for enhancing the structural feature utility. Fig. 2 shows the overall architecture of our LasUIE GLM.

LasUIE takes a three-stage training process, where a structure-aware post-training is inserted between the pre-training and fine-tuning stages for structure learning. LasUIE takes an existing well pre-trained GLM parameters (e.g., BART, T5) as initiation. During structure-aware post-training stage our GLM carries out unsupervised syntax induction based on unlabeled plain texts (cf. §4.2). Thereafter, LasUIE is fine-tuned on the in-house training data, along with which we perform task-oriented structure adaptive tuning (cf. §4.3). We also note that LasUIE takes a consistent paradigm of text-to-text generation throughout the whole three stages, which ensures a minimum information loss from the early trainings to the final predicting.

## 4.2 Unsupervised Structure-aware Post-training

**Heterogeneous structure inductor.** As cast earlier, although LMs are able to learn certain linguistic knowledge from generic pre-training, the signal strength of learned syntax is quite weak to contribute IE enough [65, 30]. In the structure-aware post-training stage, we aim to unsupervisedly enrich our GLM with sufficient structural knowledge, reinforcing the awareness of linguistic syntax.

Inspired by Shen et al. (2021) [60], we explore a heterogeneous structure inductor (HSI) stacked on top of GLM encoder to reach the above goal. HSI induces linguistic structures based on the foundation of syntax distance measurements [58]. We employ two heterogeneous syntax measurements, i.e., $O^C = \{o_1^c, \cdots, o_{n-1}^c\}$ ($o_{<1}^c = o_{>n-1}^c = \infty$) for measuring constituency syntax, and $O^D = \{o_1^d, \cdots, o_n^d\}$ for measuring dependency syntax. As illustrated in Fig. 2(b), $o_i^c$ is a real value depicting the height of the lowest common ancestor between two consecutive words $w_i$ and $w_{i+1}$; while $o_i^d$ is a real value describing the spanning distance between the words linking to $w_i$. Intuitively, bigger $o_i^c$ means bigger information divergence of the split point between the two sides of phrasal span, and larger $o_i^d$ implies wider range of connections, i.e., longer-term dependent relations. As revealed that the syntax features are best learned at lower layer of GLM encoder [23], HSI thus takes the first-layer encoding representations $\boldsymbol{h}_i^1$ as input and produce syntax context representations via convolution operation: $\boldsymbol{h}_i^* = \text{Conv}(\boldsymbol{h}_i^1)$. Based on $\boldsymbol{h}_i^*$, HSI represents $o_i^c$ and $o_i^d$ as:

$$o_i^c = \boldsymbol{V}^c \text{Tanh}(\boldsymbol{W}[\boldsymbol{h}_i^*; \boldsymbol{h}_{i+1}^*]), \quad o_i^d = \boldsymbol{V}^d \text{Tanh}(\boldsymbol{W}\boldsymbol{h}_i^*). \tag{1}$$

Then, two rules are made for generating two heterogeneous syntax based on the two measurements [60], which also helps coordinate two types of structures so that they can co-exist together and legally.

▶ **Rule** $\Gamma_C$: *A smallest constituent span $C_{[l,r]}$ of $w_i$ ($l < i < r$) should satisfy $(o_{l-1}^c > o_i^d) \& (o_r^c > o_i^d)$.* For example as in Fig. 2(b), $o_2^c(=4) > o_3^d(=3.5)$ and $o_8^c(=\infty) > o_3^d$, thus $C_{[3,8]}$ is the valid minimum span for $w_3$.

▶ **Rule** $\Gamma_D$: *Generalizing $w_i$ as a potential span $C_{[l=i,r=i]}$, the dependent head of any word in $C_{[l,r]}$ is $w_j \leftarrow argmax_{k \in [l,r]}(o_k^d)$.* For example, the maximum $o^d$ in constituent span of $C_{[3,8]}$ is $o_6^d = 4.5$, thus dependent head of the word in $C_{[3,8]}$ is $w_6$.

Based on rule $\Gamma_C$ we first generate all possible phrasal spans and organize them into a constituency tree $\mathcal{T}^C$, then constructing the dependency tree $\mathcal{T}^D$ according to rule $\Gamma_D$. We parameterize the above structure construction process so as to make it all differentiable, i.e., by describing into the probabilistic perspective. We represent the span $C_{[l,r]}$ distribution as:

$$p_c(c_k|w_i) = p(w_l|w_i) \cdot p(w_r|w_i)$$
$$= [\sigma(o_i^d - \underset{k \in [l,i)}{\text{Max}}(o_k^c)) - \sigma(o_i^d - \underset{k \in (l,i)}{\text{Max}}(o_k^c))] \cdot [\sigma(o_i^d - \underset{k \in [i,r)}{\text{Max}}(o_k^c)) - \sigma(o_i^d - \underset{k \in [i,r]}{\text{Max}}(o_k^c))], \tag{2}$$

where $c_k$ is a short hand for $C_{[l,r]}$, $\sigma$ is a sigmoid function. We then depict the rule $\Gamma_D$, and represent the word-word dependent distribution:

$$p_d(w_j|w_i) = p_d(w_j|c_k) \cdot p_c(c_k|w_i) = p_c(c_k|w_i) \cdot \exp(\boldsymbol{h}_j^L) / \sum_{k=l}^{r} \exp(\boldsymbol{h}_k^L), \tag{3}$$

where we use the top-layer encoder representation $\boldsymbol{h}_i^L$, by which we encourage the final encoding representations to learn from the low-layer syntax-rich representations. We note that the above structure induction is carried out in each of multi-head attention blocks (total $M$) in Transformer. This means that multiple distinct syntax trees of each type (i.e., $\mathcal{T}^C$ and $\mathcal{T}^D$) will be induced.

**Structural broadcaster.** It is a high chance that in GLM encoder the mainstay contextual representations will weaken the structural features and thus hurt the structure utility at decoder. To combat this, we propose a SB module, by which we explicitly collect varying trees of a type and compacted them into a forest, respectively, i.e., constituency forest $\mathcal{F}^C$ and dependency forest $\mathcal{F}^D$. According to prior studies [48, 43, 62], comparing to the optimal 1-best syntax tree, a compact forest advances in higher structure recall, which allows to learn a better bias for task. In SB, the structural priors from the syntax forests are explicitly broadcast into each decoding step for guiding the generation process.

Technically, SB selects and ranks candidate tree substructures based on the probabilistic confidence (Eq.2&3), which are then compacted into a forest based on the K-best maximum spanning tree (MST) algorithm [1, 88]. We then model the two types of forests with a graph attention model (GAT) [67] respectively, during which the decoding representation $e$ at each step is attended to spot the high-order structural feature $u$ at global level:

$$u^{c/d} = \text{GAT}(\mathcal{F}^{C/D}, e \mid h^1), \tag{4}$$

$$u = u^d \oplus u^c. \tag{5}$$

Further via a cross-attention operation (cf. Fig. 3) we navigate the encoder representation $h^L$ and the structural feature $u$ into the updated encoder representation $e^*$:

$$e^* = \text{Softmax}(\frac{h^L \cdot u}{\sqrt{d}}) \cdot e. \tag{6}$$

**Post-training objectives.** The first objective is performing seq-to-seq style language modeling, i.e., 'corrupting + reconstructing' the inputs [32], which is identical to the pre-training objectives. We denote the language modeling loss as $\mathcal{L}_W$.

Along with the language modeling we then promote the unsupervised structure induction, including the one for dependency syntax and the one for constituency syntax:

$$\mathcal{L}_D = -\sum_m^M \sum_i^n \sum_j^n \log p_d(w_j|w_i), \tag{7}$$

$$\mathcal{L}_C = -\sum_m^M \sum_i^n \sum_k^K [\log p_c(c_k|w_i) + \log(\exp \phi(c_k)/\mathcal{Z})], \tag{8}$$

where $\phi(c_k) = \frac{(h^c)^T \cdot h^L_{[l,r]}}{||h^c|| \cdot ||h^L_{[l,r]}||}$ is a span similarity score between the constituent phrase $c_k$ and the counterpart text span derived from the top-layer encoder. $\mathcal{Z}$ is for normalization.

We also perform structure diversifying regularization (SDR), putting constraints on the varying trees induced from different multi-head encoder attentions so as to ensure structure diversification.

$$\mathcal{L}_{SDR} = -\sum_m^M \sum_k^M ||A_m \odot A_k||, \quad k \neq m, \tag{9}$$

where $A_m$ or $A_k$ is an attention map. We put all the above objectives together as the post-training target: $\mathcal{L}_{PRT} = \mathcal{L}_W + \mathcal{L}_D + \mathcal{L}_C + \mathcal{L}_{SDR}$.

### 4.3 Task-oriented Structure Fine-tuning

After structure-aware post-training, our GLM is finally fine-tuned on a specific terminal IE task to learn the on-demand features. This target is an empirical risk minimization with cross-entropy loss:

$$\mathcal{L}_{Task} = -\sum^D \log p(y|x), \quad (10)$$

where $D$ is the mini-batch size.

Meanwhile, we perform further task-oriented structural fine-tuning, adapting the learned structure information to the task-specific IE structures, for example, in dependency structure pruning those trivial word-word connections and adjusting the range

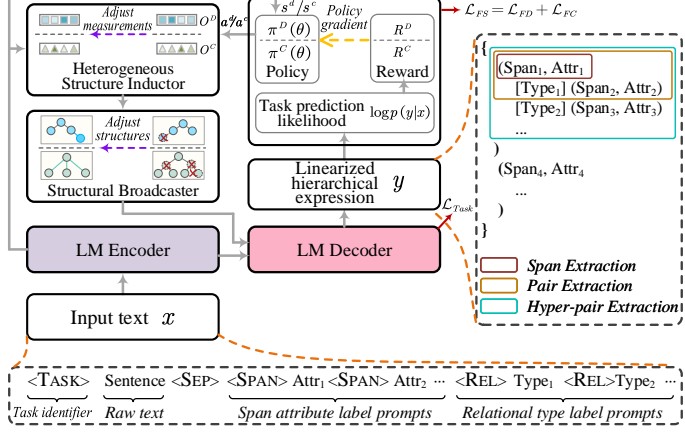

Figure 3: Fine-tuning our GLM with structure adaptive learning.

of dependent paths; in constituency structure refining the phrase widths and granularities. Our main idea is to amend the syntax attribute (i.e., dependency links and constituent compositions) by directly taking the feedback of end task performance. Therefore, we employ the stochastic policy gradient algorithm [76]. As shown in Fig. 3, the actions $a_i^{c/d} \in (-1, 1)$ are real values sampled with probabilities from a Gaussian distribution, by which we maintain a continuous control over syntax measurements, i.e., $O^D$ and $O^C$.

$$\bar{a}_i^{c/d} \sim \pi^{C/D}(\boldsymbol{s}_i^{c/d}; \theta^{c/d}) = \mathcal{N}(0, I) \,, \tag{11}$$

$$a_i^{c/d} = 2\sigma(\bar{a}_i^{c/d}) - 1 \,, \tag{12}$$

$$o_i^{c/d} := a^{c/d} + o_i^{c/d} \,, \tag{13}$$

where the $\boldsymbol{s}_i^{c/d} = (\boldsymbol{h}_i^1 \oplus \boldsymbol{h}_i^* \oplus \boldsymbol{h}_i^L)$ are the state representations as the inputs of the policy agents. The policy agents $\pi^C(\theta^c)$ and $\pi^D(\theta^d)$ are two parameterized two-layer feedforward networks, respectively. We design the reward of the policy as the probability of correct task prediction, such that the structure adjustments are directly supervised by terminal task's signals: $R^{C/D} = \log p(y|x)$. The learning target of each policy is to maximize the corresponding expected reward:

$$\mathcal{L}_{FD} = - \sum_{(a_1^d s_1^d \cdots a_n^d s_n^d)} \prod_i p(a_i^d | \boldsymbol{s}_i^d; \theta^d) \cdot R_i^D \,, \tag{14}$$

$$\mathcal{L}_{FC} = - \sum_{(a_1^c s_1^c \cdots a_n^c s_n^c)} \prod_i p(a_i^c | \boldsymbol{s}_i^c; \theta^c) \cdot R_i^C \,. \tag{15}$$

We summarize all the fine-tuning targets: $\mathcal{L}_{FT} = \mathcal{L}_{Task} + \mathcal{L}_{FS}$, where $\mathcal{L}_{FS} = \mathcal{L}_{FD} + \mathcal{L}_{FC}$.

## 5 Experiments

### 5.1 Setups

We take the pre-trained T5 Base as default backbone GLM. We use the plain texts from *Wikipedia*[2] and *BooksCorpus*[3] corpora for the post-training. To cover all three UIE prototypes, we consider 7 representative IE tasks with corresponding data: 1) NER: CoNLL03 [66], OntoNote [53], ACE04 [11], ACE05 [22]; 2) RE: CoNLL04 [57], NYT [56], ACE05 [22]; 3) AOP: Res14 [51]; 4) ASTE: Res14 [51]; 5) ORL: MPQA [74]; 6) SRL: CoNLL12 [53]; 7) EE: ACE05 [22]. Each dataset has its own split, and we follow the same practice of the relevant prior works when using it.

We verify the IE performances under the traditional separate scheme and the recent unified scheme, respectively. *1) In separate IE*, we compare with the current SoTA systems (all using Large version LM/GLM) of each specific data; meanwhile we implement a T5 (Base version) system, namely GEN-T5, using the same generative manner (based on prompt input, generating LHE as ours) for running each task individually. We also retrofit the GEN-T5 system by injecting into the external syntax parse trees via additional training on the syntax annotated corpus, including the dependency syntax (+*DepSyn*), constituency syntax (+*ConSyn*) and both two types syntax (+*Dep&ConSyn*), respectively. *2) In unified IE*, we mainly make comparisons with the current UIE system [42]. Note that the default UIE model (marked as UIE$^{*\dagger}$) in raw paper uses T5 Large and meanwhile takes additional supervised pre-training on the large-scale IE corpus (the version without supervised IE pre-training marked as UIE$^*$). To ensure fair comparisons, we re-implement their system with T5-Base parameters and without supervised IE pre-training, marked as UIE. Same as to GEN-T5, we also retrofit the UIE model by integrating heterogeneous syntax parse trees in different combinations.

Following each of previous works, we use the F1 evaluation metrics. For each task, we consider the end-to-end prediction. For example, for the span extraction (NER), we measure if both the mention span and the mention attribute are correct. For the pair(/hyper-pair) extraction, we measure if the span boundary & span attribute & relation & type are all correct simultaneously.

### 5.2 Main Results

We present the overall comparison results on various IE tasks in Table 1 and Table 2 under the fully-supervised and low-resource scenario, respectively. As can be seen, our proposed LasUIE

---

[2]https://autonlp.ai/datasets/wikipedia-news-corpus
[3]https://huggingface.co/datasets/bookcorpus

Table 1: Overall IE performances by different methods (all using LM/GLM). Models with ∗ (M1, M6, M7 & M8) refers to the use of Large version LM, where scores by M1, M6 & M7 are copied from their raw paper [42]. UIE*† (M6) takes additional supervised pre-training on the large-scale IE corpus. **Bold**: the best results among the comparisons using Large and Base LMs, respectively.

| Task&Data | Span Extraction | | | | Pair Extraction | | | | | Hyper-pair Extraction | | | Avg. |
|---|---|---|---|---|---|---|---|---|---|---|---|---|---|
| | NER | | | | RE | | | AOP | ASTE | ORL | SRL | EE | |
| | CoNLL03 | OntoNote | ACE04 | ACE05 | CoNLL04 | NYT | ACE05 | Res14 | Res14 | MPQA | CoNLL12 | ACE05 | |
| ● *Separate IE* | | | | | | | | | | | | | |
| M1  SoTA* | 93.2 | 91.9 | 86.8 | 84.7 | 73.6 | 92.7 | 65.6 | 69.3 | 73.6 | 53.0 | 73.5 | 48.3 | 75.5 |
| M2  GEN-T5 | 91.0 | 89.1 | 84.3 | 83.0 | 69.4 | 90.3 | 60.2 | 62.5 | 71.8 | 49.8 | 69.3 | 43.7 | 72.0 |
| M3    +DepSyn | 91.5 | 89.5 | 84.9 | 83.4 | 70.3 | 91.8 | 62.4 | 64.3 | 72.6 | 51.5 | 70.8 | 45.5 | 73.2 |
| M4    +ConSyn | 92.1 | 90.0 | 85.3 | 83.8 | 69.8 | 90.9 | 61.5 | 63.1 | 72.3 | 50.7 | 70.1 | 44.3 | 72.8 |
| M5    +Dep&ConSyn | 92.3 | 90.4 | 85.3 | 84.0 | 71.2 | 92.1 | 63.3 | 66.0 | 73.0 | 51.8 | 71.3 | 46.2 | 73.9 |
| ● *Unified IE* | | | | | | | | | | | | | |
| M6  UIE*† | 93.0 | / | **86.9** | 85.8 | 75.0 | / | 66.0 | / | 74.5 | / | / | / | / |
| M7  UIE* | 92.1 | / | 86.5 | 85.5 | 73.1 | 93.5 | 64.7 | / | / | / | / | / | / |
| M8  **LasUIE*** (Ours) | **93.2** | **93.0** | 86.8 | **86.0** | **75.3** | **94.2** | **66.4** | **73.6** | **75.2** | **57.8** | **76.3** | **51.7** | **77.4** |
| M9  UIE | 91.4 | 89.7 | 85.0 | 83.5 | 70.5 | 91.0 | 61.6 | 65.8 | 72.8 | 50.8 | 70.2 | 44.6 | 73.1 |
| M10    +DepSyn | 91.8 | 90.0 | 85.3 | 83.7 | 71.2 | 92.0 | 62.9 | 67.6 | 73.5 | 52.0 | 71.5 | 46.4 | 74.0 |
| M11    +ConSyn | 92.0 | 90.5 | 85.6 | 84.0 | 70.8 | 91.3 | 62.1 | 66.1 | 73.1 | 51.3 | 71.0 | 45.2 | 73.6 |
| M12    +Dep&ConSyn | 92.3 | 90.7 | 85.8 | 84.5 | 71.7 | 92.4 | 63.4 | 68.2 | 73.7 | 53.6 | 72.6 | 47.0 | 74.6 |
| M13  **LasUIE** (Ours) | **92.6** | **92.0** | **86.3** | **85.0** | **73.2** | **93.0** | **64.4** | **70.2** | **74.8** | **56.0** | **74.7** | **49.0** | **75.9** |
| M14    w/o SB | 92.0 | 90.7 | 85.5 | 84.2 | 71.5 | 91.8 | 62.9 | 68.3 | 73.4 | 54.7 | 73.4 | 47.7 | 74.6 |
| M15    w/o $\mathcal{L}_{SDR}$ | 92.2 | 91.6 | 86.2 | 84.8 | 72.8 | 92.4 | 64.1 | 70.0 | 74.4 | 55.5 | 74.0 | 48.6 | 75.6 |
| M16    w/o $\mathcal{L}_{FS}$ | 92.4 | 91.4 | 85.9 | 84.7 | 71.8 | 92.0 | 63.6 | 69.1 | 73.6 | 54.2 | 73.0 | 47.1 | 74.9 |

framework consistently outperforms the baseline UIE and other SoTA models on all tasks in both two learning scenarios, under both the Large or Base T5 initiations. This demonstrates the efficacy of our proposal. Also we compare the counterparts between M2-M5 and M9-M12, where the only difference between these generative methods lies in the seprate or unified modeling of IE. From the results we learn that the unified modeling of IE (i.e., UIE) is more effective than the traditional separate modeling of specific IE task. This verifies that the universal modeling helps share the task-invariant IE features, coinciding with the findings in [42].

## 5.3   In-depth Analysis

To aid better understanding the strengths of our method, we further present in-depth analyses from varying angles, i.e., by asking four key questions concentrating on the structure-aware GLM for UIE.

**Q1: Can fusing syntax structure knowledge into GLM contribute to UIE?**    Let's compare the results in Table 1: M2 vs. M3&M4&M5 in separate IE setup, and M9 vs. M10&M11&M12 in unified IE setup, where *either in separate or unified IE setup, integrating additional linguistic syntax features into GLM evidently improves all end task performances.* Interestingly, different tasks can receive varying degree of improvements from the syntax features. Importantly, we see in Table 2 that with the aids of structure knowledge, the performances of low-resource transfer can be promoted, especially in the combination with the unified modeling of IE tasks. This proves that *the syntactic structures in GLM can serve as IE task-invariant features, further contributing to UIE.*

**Q2: What are the differences to integrate the constituency and dependency syntactic structure?** We now observe the results of different tasks in both Table 1&2, and we can find that *on the span extraction type IE (i.e., NER) the improvements from constituency syntax prevail, while the dependency type of structure features dominate the pair-wise tasks, i.e., (hyper-)pair extraction.* We further analyze the error rate on the predictions of two kernel elements of IE, i.e., boundary recognition and relation detection, respectively on various tasks. We see from Fig. 4 that *the constituency structure more tends to offer key clues for the boundary recognition; while the dependent trees are more apt to cope with the relation detection, solving long-range dependence issue.* This shows that two heterogeneous structures have complementary advantages to UIE. *Thus, when combining both of them together, all the end tasks receive the enhancements to the greatest extent.*

Table 2: Performances on low-resource settings by IE models (using T5 Base). The scores by UIE$^\dagger$, which takes additional pre-training with large-scale supervised IE corpus, are copied from raw paper [42]. GEN-5 model takes separate IE modeling on each task, while the other models take unified IE.

| Task&Data | Span Extraction | | | | Pair Extraction | | | | | Hyper-pair Extraction | | | Avg. |
|---|---|---|---|---|---|---|---|---|---|---|---|---|---|
| | NER | | | | RE | | | AOP | ASTE | ORL | SRL | EE | |
| | CoNLL03 | OntoNote | ACE04 | ACE05 | CoNLL04 | NYT | ACE05 | Res14 | Res14 | MPQA | CoNLL12 | ACE05 | |
| **● 1-shot** | | | | | | | | | | | | | |
| UIE$^\dagger$ | **46.4** | / | / | / | 22.1 | / | / | / | / | / | / | / | / |
| GEN-T5+*Dep&ConSyn* | 27.2 | 20.4 | 14.8 | 17.6 | 8.2 | 25.7 | 10.8 | 12.8 | 10.8 | 1.1 | 6.5 | 1.5 | 13.1 |
| UIE+*Dep&ConSyn* | 30.3 | 23.6 | 17.5 | 20.7 | 12.8 | 26.7 | 14.3 | 16.7 | 13.0 | 2.8 | 14.0 | 3.8 | 16.4 |
| **LasUIE** | 39.4 | **47.6** | **38.5** | **44.7** | **25.7** | **45.0** | **26.7** | **30.0** | **38.4** | **18.9** | **32.8** | **23.7** | **34.3** |
| **● 10-shot** | | | | | | | | | | | | | |
| UIE$^\dagger$ | 73.9 | / | / | / | 52.4 | / | / | / | / | / | / | / | / |
| GEN-T5+*Dep&ConSyn* | 67.4 | 64.7 | 49.2 | 52.8 | 45.6 | 50.8 | 37.4 | 19.7 | 17.8 | 5.4 | 18.7 | 12.2 | 36.8 |
| UIE+*Dep&ConSyn* | 69.5 | 68.4 | 52.8 | 54.1 | 51.8 | 56.0 | 43.8 | 22.5 | 26.1 | 10.5 | 23.2 | 17.6 | 41.4 |
| **LasUIE** | **74.0** | **78.3** | **60.3** | **65.3** | **55.0** | **67.7** | **46.1** | **42.4** | **48.8** | **25.4** | **45.8** | **27.1** | **53.0** |
| **● 1% data** | | | | | | | | | | | | | |
| UIE$^\dagger$ | **82.8** | / | / | / | 30.8 | / | / | / | / | / | / | / | / |
| GEN-T5+*Dep&ConSyn* | 79.5 | 72.4 | 58.3 | 61.7 | 17.8 | 35.8 | 15.4 | 15.3 | 15.3 | 3.3 | 10.7 | 3.4 | 32.4 |
| UIE+*Dep&ConSyn* | 80.6 | 73.2 | 60.4 | 63.8 | 23.5 | 40.4 | 22.7 | 20.6 | 18.5 | 5.3 | 17.6 | 10.2 | 36.4 |
| **LasUIE** | 82.1 | **84.5** | **65.7** | **70.1** | **32.0** | **53.6** | **34.2** | **34.8** | **41.7** | **21.0** | **39.8** | **25.7** | **48.8** |
| **● 10% data** | | | | | | | | | | | | | |
| UIE$^\dagger$ | 89.6 | / | / | / | 59.2 | / | / | / | / | / | / | / | / |
| GEN-T5+*Dep&ConSyn* | 89.0 | 84.0 | 71.3 | 68.8 | 52.4 | 80.4 | 45.7 | 56.0 | 59.7 | 22.4 | 50.7 | 26.7 | 58.9 |
| UIE+*Dep&ConSyn* | 89.3 | 85.8 | 72.1 | 70.6 | 54.9 | 82.5 | 47.6 | 58.3 | 62.6 | 27.4 | 54.3 | 31.7 | 64.4 |
| **LasUIE** | **91.6** | **89.3** | **83.6** | **81.7** | **60.8** | **86.0** | **50.5** | **63.0** | **66.7** | **36.0** | **58.4** | **38.4** | **67.2** |

Figure 4: Error rates on boundary recognition and relation detection, respectively.

**Q3: For UIE, is it more advanced for GLM to automatically learn latent structures than injecting external syntax parse trees?** First, the comparisons in the main results directly prove the advance of using latent structural features for UIE. For example, under the fair comparison, our LasUIE beats the UIE+*Dep&ConSyn* model with average 1.3%(=75.9-74.6) F1 improvement. Even comparing with the UIE*$^\dagger$ that takes additional pre-training on large-scale supervised IE corpus, LasUIE keeps its superiority in almost all cases. This evidently verifies that *it is necessary for LMs to automatically learn latent structure information for better UIE.* The underlying reason of our model's improvements could be that *the dynamically learned richer structural knowledge in LasUIE largely avoids the noises that are introduced in external syntax parse annotations.* Besides, as shown in Fig. 4, LasUIE reduces the errors on predicting the mention boundaries and relational pairings more significantly than the baseline counterparts.

Further we step into our LasUIE system itself, and inspect the ablation models, M14-M15, as shown in Table 1. We see that the proposed structural broadcaster module plays important role to the overall system, i.e., without SB, LasUIE is downgraded to the level of UIE+*Dep&ConSyn*. Also the structure diversification regularization mechanism serves positive effect.

**Q4: Is it necessary to further fine-tune the structures in GLM for UIE?** According to the results of the ablation model, M16, in Table 1, we can directly claim the answer is positive. Without performing structural fine-tuning, the results by LasUIE hurt clearly, with averaged 1.0%(=75.9-74.9) F1 drop. We next dig into the structural fine-tuning mechanism, analyzing how the auto-induced structural features influence the UIE performances.

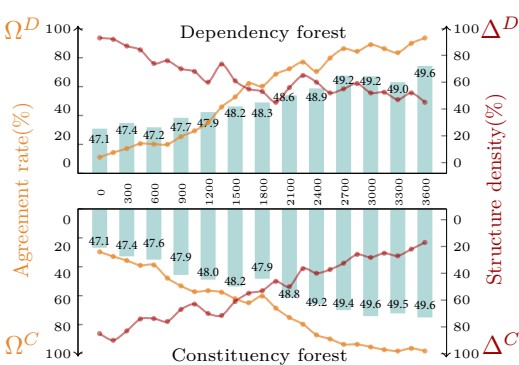
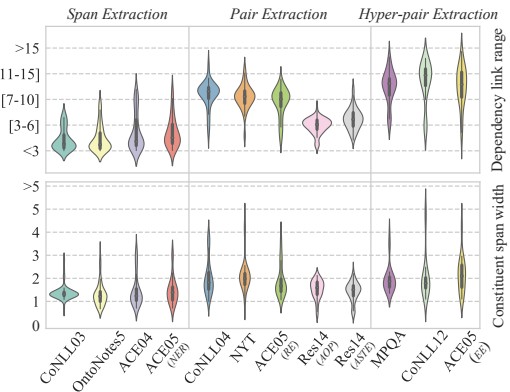

Figure 5: Trajectories of the changing structure agreement rates and densities during task-oriented structure fine-tuning, based on event extraction (ACE05). $X$-axis is the iteration steps for fine-tuning. Bars means the task performances (F1).

Figure 6: The distributions of the range of word-word dependency link (words) in forest $\mathcal{F}^D$ and the constituency phrasal span width (words) in forest $\mathcal{F}^C$ on each data.

We first study the changing trajectories of the 1) the structure agreement rates $\Omega^{D/C}$ and 2) the structure densities $\Delta^{D/C}$ of the structural forests $\mathcal{F}^{D/C}$, during fine-tuning. $\Omega^C$ or $\Omega^D$ is defined as the percentage that gold spans correspond to the phrasal spans in the constituency forest $\mathcal{F}^C$, or the gold relational pairs coincide with the word-word edges in the dependency forest $\mathcal{F}^D$. As plotted in Fig. 5, along with the fine-tuning process the task performance climbs gradually. Meanwhile, both the agreement rate $\Omega^{D/C}$ increases, which means that *the structural fine-tuning indeed can effectively adjust the learned structural information towards task-specific*. Also, the structural densities of two forests change from dense to sparse, which depicts a structure pruning process in our system.

Finally, in Fig. 6 we present the distribution of the range of word-word dependency links in $\mathcal{F}^D$ and distribution of the constituency phrasal span width in $\mathcal{F}^C$. We can discover that different end tasks rely on subtly varying structural features or attributes. For example, hyper-pair extraction tasks require longer-range dependency features for relation determination, comparing to the IE tasks of other prototypes. In turn, this certifies that *our system can correctly learn the peculiar structural bias for a specific IE task*, thanks to the task-oriented structure fine-tuning mechanism.

## 6 Conclusion and Discussion

This work investigates a novel structure-aware generative language model (GLM) that learns rich heterogeneous syntactic structure representations for better unified information extraction (UIE). First, a well pre-trained GLM is taken as backbone to reach the goal of UIE, feeding with label prompt-based texts and predicting linearized hierarchical expressions that describe the actual IE target. During post-training, the proposed heterogeneous structure inductor automatically generates rich structure information without relying on any additional syntax annotation. A structural broadcaster then compacts various trees into forests for enhancing the structural feature utility and guiding better context generation. The learned structural knowledge is further fine-tuned on the in-house training data so as to adapt into the task-specific need. Extensive experiments and in-depth analyses demonstrate the efficacy of our system on improving the UIE.

**Potential impact and limitations of the work.** The proposed structure-aware GLM learns syntactic knowledge relies only on the plain texts with easy access, without any cost of large-scale human-labor annotations. The system will benefit the development of IE community, i.e., training one single unified model for effectively solving various IE tasks, which especially addresses the issue of IE data annotation scarcity in the real-life applications. One biggest potential risk is that the GPU-based training of our language model will cost energy consumption and $CO_2$ emissions.

## Acknowledgments and Disclosure of Funding

This research is supported by the Sea-NExT Joint Lab. We would also like to thank the anonymous reviewers for their valuable feedbacks.

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
