# A  Detail of Syntax Structure Induction

Here we unfold the induction of Eq. (2) for the structure induction. Based on the rule $\Gamma_C$ we have:

$$p(o_i^d > o_k^c) = \sigma(o_i^d > o_k^c).$$

Then, the probability that the $l$-th ($l < i$) token is inside $C_{[l,r]}$ is equal to the probability that $o_i^d$ is larger then the maximum $o_k^c$ between $l$ and $i$:

$$p(l \in C_{[l,r]}) = p(o_i^d > \text{Max}(o_{i-1}^c, \cdots, o_l^c)) = \sigma(o_i^d - \underset{k\in[l,i)}{\text{Max}}(o_k^c)).$$

Thus, the conditional probabilistic of $p(w_l|w_i)$ becomes:

$$p(w_l|w_i) = \sigma(o_i^d - \underset{k\in[l,i)}{\text{Max}}(o_k^c)) - \sigma(o_i^d - \underset{k\in(l,i)}{\text{Max}}(o_k^c)).$$

Similarly, for $p(w_r|w_i)$:

$$p(w_r|w_i) = \sigma(o_i^d - \underset{k\in[i,r)}{\text{Max}}(o_k^c)) - \sigma(o_i^d - \underset{k\in[i,r]}{\text{Max}}(o_k^c)).$$

And finally, we can derive the probability of phrasal span $C_{[l,r]}$:

$$p_c(c_k|w_i) = p(w_l|w_i) \cdot p(w_r|w_i)$$
$$= [\sigma(o_i^d - \underset{k\in[l,i)}{\text{Max}}(o_k^c)) - \sigma(o_i^d - \underset{k\in(l,i)}{\text{Max}}(o_k^c))] \cdot [\sigma(o_i^d - \underset{k\in[i,r)}{\text{Max}}(o_k^c)) - \sigma(o_i^d - \underset{k\in[i,r]}{\text{Max}}(o_k^c))].$$

# B  Baseline Specification

In our experiments we take the current SoTA methods as the separate IE comparisons on each specific task and data. Here Table 3 shows the specifications of these baseline models. Parts of the results are directly copied from their raw papers, where the Large version LM or GLM is used, while part of the results are from our reimplementation.

Table 3: SoTA baseline systems for different IE tasks.

| Task | Dataset | Model | LM Type | Result Source |
|------|---------|-------|---------|---------------|
| NER | CoNLL03 | Wang et al. [70] | RoBERTa-Large | Raw paper |
| | ACE04/05 | Yan et al. [80] | BART-Large | Raw paper |
| | OntoNote | Li et al. [35] | BERT-Large | Reimplementation |
| RE | CoNLL04 | Wang and Lu [69] | ALBERT-large | Raw paper |
| | NYT | Zheng et al. [86] | BERT-Large | Raw paper |
| | ACE05 | Zhong and Chen [87] | ALBERT-XXLarge | Raw paper |
| AOP | Res14 | Wu et al. [77] | BERT-Large | Reimplementation |
| ASTE | Res14 | Zhang et al. [84] | BERT-Large | Reimplementation |
| ORL | MPQA | Wu et al. [78] | BERT-Large | Reimplementation |
| SRL | CoNLL12 | Fei et al. [17] | RoBERTa-Large | Reimplementation |
| EE | ACE05 | Lin et al. [40] | BERT-Large | Reimplementation |

# C  Task Label Prompts

Each task with each dataset will come with different task label, including the *span attribute labels* and *relation type labels*, which will be used as the label prompts in input. Table 4 summarizes all the label prompts of each dataset. Also we note that, instead of directly taking the raw label abbreviations as label prompts, we use the full names of labels in natural languages, such that we can fully utilize their semantic representations in GLM.

Table 4: The span attribute labels and relation type labels of different tasks and datasets for building the task labels prompts. We note that we replace the raw label abbreviations with their full names in natural languages, so as to fully utilize their semantic representations in GLM.

| Task | Dataset | Span attribute labels | Relation type labels |
|---|---|---|---|
| NER | CoNLL03 | location, organization, person, miscellaneous | / |
| | OntoNote | person, nationality, facility, organization, geographical political, location, product, event, work of art, law, language, time, date, percent, money, quantity, ordinal, cardinal | / |
| | ACE04 | person, organization, location, facility, geographical political, vehicle, weapon | / |
| | ACE05 | person, organization, location, facility, geographical political, vehicle, weapon | / |
| RE | CoNLL04 | location, organization, people, other | kill, live in, located in, organization in, work for |
| | NYT | location, organization, person | administrative divisions, advisors, capital, children, company, contains, country, ethnicity, founders, geographic, distribution, industry, location, major shareholder of, major shareholders, nationality, neighborhood of, people, place founded, place lived, place of birth, place of death, profession, religion, teams |
| | ACE05 | person, organization, location, facility, geographical political, vehicle, weapon | agent artifact, general affiliation, organization affiliation, part whole, personal social, physical |
| AOP | Res14 | aspect, opinion | default relation |
| ASTE | Res14 | aspect, opinion | positive, neutral, negative |
| ORL | MPQA | opinion, role | holder, target |
| SRL | CoNLL12 | default argument, default predicate | agent (ARG0), patient (ARG1), instrument and end state (ARG2), starting point and benefactive and attribute (ARG3), ending point (ARG4), direction (DIR), location (LOC), manner (MNR), extent (EXT), reciprocals (REC), secondary predication (PRD), purpose (PNC), cause (CAU), discourse and connectives (DIS), adverbial and general purpose (ADV), modal verb (MOD), negation (NEG), time (TMP) |
| EE | ACE05 | default argument, acquit, appeal, arrest jail, attack, born, charge indict, convict, declare bankruptcy, demonstrate, die, divorce, elect, end organization, end, position, execute, extradite, fine, injure, marry, meet, merge organization, nominate, pardon, phone write, release parole, sentence, start organization, start position, sue, transfer money, transfer, ownership, transport, trial hearing | person, agent, victim, instrument, attacker, target, instrument, time, place, artifact, vehicle, price, origin, destination, time, buyer, seller, beneficiary, giver, recipient, beneficiary, org, money, entity, position, crime, defendant, prosecutor, adjudicator, plaintiff, sentence |