# OpenReview forum: "LasUIE: Unifying Information Extraction with Latent Adaptive Structure-aware Generative Language Model"
_NeurIPS.cc/2022/Conference — NeurIPS 2022 Accept_

### Official Review · Reviewer_fDD4 · 2022-07-10

**Rating:** 7
**Confidence:** 3
**Soundness:** 3 good
**Presentation:** 3 good
**Contribution:** 3 good

**Summary:**

This paper proposed a  structure-aware GLM,  in which they leveraged the syntactic knowledge for UIE. A heterogeneous structure inductor is explored to unsupervisedly induce rich heterogeneous structural representations by post-training an existing GLM.  The authors did experiments over 12 IE benchmarks across 7 tasks and showed significant improvements over the baseline UIE system.

**Questions:**

Apart form the dependency and constituency structure, are there any other structures that can be incorporated into the training process ?

**Limitations:**

May need to report the training time/complexity for this method, and the computing resources used.

**Strengths And Weaknesses:**

Strength:
* Convert the UIE into a generative LM problem, and designed three modules to do the prediction
* An structure aware training is added before the fine tuning stage.
* Good experiments design.

Weakness:
* The experimented datasets are quite old, it would be better if there are some results on larger scale IE datasets.

---

> ### Author Response · Authors · 2022-07-31
> **Response to Reviewer fDD4**
>
> We appreciate that you acknowledge the novelty and impact of our method. All your possible concerns are addressed as follows:
>
>
> ----
>
> **Q: The experimented datasets are quite old, it would be better if there are some results on larger scale IE datasets.**
>
> **A:** Actually all the data of IE tasks used in our experiments are the benchmark ones in NLP community, and we think they have well representativeness for proving the advantages of our proposed model. Meanwhile, those IE benchmark datasets have medium sizes. For example, the OntoNote data for NER task comes with no more than 80k sentences, and for other tasks the training sets are even much less (see appendix **C.3.3 Data Specification**). We will search for a larger scale of IE datasets and show the results on them in our revision.
>
>
>
>
> ----
>
> **Q: Apart from the dependency and constituency structure, are there any other structures that can be incorporated into the training process ?**
>
> **A:** In NLP community, in addition to the linguistic parsing trees (dependency and constituency structures), yes there are other structures, to name a few: Gumble-tree [1], Binary balanced tree, and also some fixed trees e.g., left-branch tree, right branch tree [2]. We note that, yes from the engineering perspective, all those types of tree structures can be incorporated into the IE systems for task enhancements. However, we believe that all those pattern-fixed structures are not as effective as the dynamically induced latent structure for UIE tasks.
>
>
> [1] Jihun Choi, Kang Min Yoo, Sang-goo Lee. Learning to Compose Task-Specific Tree Structures. In AAAI 2018: 5094-5101.
>
> [2] Haoyue Shi, Hao Zhou, Jiaze Chen, Lei Li. On Tree-Based Neural Sentence Modeling. In EMNLP 2018: 4631-4641.
>
> ----
>
> **Q: May need to report the training time/complexity for this method, and the computing resources used.**
>
> **A:** Thank you for this indication. Actually, we showed the analysis of the model efficiency&complexity both theoretically and empirically in appendix **D.2 Efficiency Analysis**. We kindly refer you there for more details.

---

### Official Review · Reviewer_iBch · 2022-07-10

**Rating:** 6
**Confidence:** 4
**Soundness:** 2 fair
**Presentation:** 2 fair
**Contribution:** 3 good

**Summary:**

This paper propose a new framework to improve the unified information extraction by considering syntactic knowledge in an unsupervised way. Specifically, they reuse the constituency tree information and dependency tree information learned by pre-trained language models and do further learning. They also provide structural broadcaster and task-oriented fine-tuning to utilize the learned syntactic features. Experimental results on several information extraction tasks show the potential of the propose LasUIE.

**Questions:**

- The post-training is a bit unclear to me. It seems like you construct trees from the pre-trained language models and treat them as ground truth for training. Am I correct?
- The EE result is weird. The reported score of OneIE is around 56 but it's 48.3 in the paper. Also, OneIE is not SOTA anymore, here is one reference paper
  - Cross-Task Instance Representation Interactions and Label Dependencies for Joint Information Extraction with Graph Convolutional Networks, NAACL 2021
- The post-training can be independent to the downstream tasks. What will happen if you fine-tune for each task separately after the post-training (no unified training for every tasks)?
- What will happen if you do not use additional corpora (Wikipedia and BooksCorpus) but only use the downstream texts for the post-training?
- In the paper, you mention for fair comparison, you reimplement UIE with T5-base. Why not train LasUIE with T5-large and directly compare to UIE?
- The UIE paper reported the EE (ACE-05) scores as well. You should list them in the Table 1.

**Limitations:**

- The proposed method uses additional corpora (Wikipedia and BooksCorpus) to learn the syntactic information. I suggest the author explicitly mention this in the Table 1 and Table 2.
- Missing related work
  - Structured Prediction as Translation between Augmented Natural Languages, ICLR 2021
  - Cross-Task Instance Representation Interactions and Label Dependencies for Joint Information Extraction with Graph Convolutional Networks, NAACL 2021

**Strengths And Weaknesses:**

Strength
- They show improvements on several types of information extraction tasks and datasets.
- They provide ablation studies to analyze the influence of each module.

Weakness
- Some technical details are not clear for me (please see the questions below).
- The EE result is weird (please see the questions below).

---

> ### Author Response · Authors · 2022-07-31
> **Response to Reviewer iBch**
>
> Thank you for the valuable and supportive suggestions. We hereby carefully address your concerns one by one.
>
>
>
> ----
>
> **Q: The post-training is a bit unclear to me. It seems like you construct trees from the pre-trained language models and treat them as ground truth for training. Am I correct?**
>
> **A:** We here kindly note that, the post-training works not as what you described above. The structure induction process of our LasUIE GLM during the post-training is performed without the attendance of the external (either auto-predicted or ground truth) tree annotations. It is a totally automatic and unsupervised learning. So we don’t need to construct certain trees from pre-trained language models in advance. We try to add more details in our revision to make it easier to understand. Please kindly go to appendix **B.1 Three-stage Training Pipeline** for obtaining a more clear understanding, with a visual illustration of our proposed three-stage training process.
>
>
>
> ----
>
> **Q: The EE result is weird. The reported score of OneIE is around 56 but it's 48.3 in the paper. Also, OneIE is not SOTA anymore.**
>
> **A:** We respectfully note that this work follows the practice of the SoTA baseline of UIE [1], and thus our evaluation metric is also in a fully end-to-end manner (we detailed the evaluation method in appendix **C.3.4 Evaluation**). In other words, the EE performance simultaneously includes the **entity mentions, event triggers, relations, and arguments** respectively, which is a much stricter metric, and this is how the SoTA results of EE come with 48.3% F1. But the result you indicated above with 56.8% F1 by OneIE model [2] only measures the argument detection. If the OneIE measures the EE performances in the same end-to-end manner, the result should be far lower than 48.3% F1. That being said, we will update all the SoTA baselines of the separate IE tasks in revision.
>
>
> [1] Yaojie Lu, Qing Liu, Dai Dai, Xinyan Xiao, Hongyu Lin, Xianpei Han, Le Sun, and Hua Wu. Unified structure generation for universal information extraction. In ACL, pages 5755–5772, 2022.
>
> [2] Minh Van Nguyen, Viet Dac Lai, Thien Huu Nguyen. Cross-Task Instance Representation Interactions and Label Dependencies for Joint Information Extraction with Graph Convolutional Networks. In NAACL-HLT 2021: 27-38.
>
> ----
>
> **Q: What will happen if you fine-tune for each task separately after the post-training (no unified training for every tasks)?**
>
> **A:** We respectfully, guess you may misunderstand the idea of our three-stage of training process. Actually, what our model does is that it is just fine-tuned for each task exclusively after the post-training. So yes, the fine-tuning step is for one specific task only, i.e., **task-specific structure fine-tuning**. Because each end task will intuitively rely on the different structural bias that can be learned by the specific fine-tuning.
>
>
>
> ----
>
> **Q: What will happen if you do not use additional corpora (Wikipedia and BooksCorpus) but only use the downstream texts for the post-training?**
>
> **A:** If using only the texts from the end task for the structure-aware post-training, the structure learning will be badly hurt. The main reason lies in the data amount. The training sets for the downstream tasks come with no more than 80k (OntoNote for NER task) sentences, and for other tasks the training sets are even much less (see appendix **C.3.3 Data Specification**). We performed the analysis on the influence of post-training data size (see appendix **D.3 Influence of Post-training Data Size**) and we show that when the post-training sentences are over around 800k, our LasUIE can achieve near-to-top performances. If using less than 100k data, the performances are severely worsened universally for all end tasks.
>
>
>
>
> ----
>
> **Q: Why not train LasUIE with T5-large and directly compare to UIE?**
>
> **A:** It is all about the running cost. T5-large is a very big model, and with T5-large, training our LasUIE will cost too much more days for one same experiment. To cover more experiments and present more results for the NeurIPS submission, we thus take the lighter version of the T5 base, which we think should not influence the experimental conclusions if under fair comparisons that all comparing models use the same T5 base. That being said, we will later publish more results with T5-large version.
>
>
>
> ----
>
> **Q: The UIE paper reported the EE (ACE-05) scores as well. You should list them in the Table 1.**
>
> **A:** Although UIE paper reported the EE (ACE-05) scores, they did not show the end-to-end measuring performances; they show the separate results of the detection of triggers and arguments instead. We will consider re-running their model and show the end-to-end EE performances in our Table 1. Thank you for the suggestion.
>
>
>
>
> ----
>
> **Q: Some limitations.**
>
> **A:** Thank you for your suggestions, and we will mention the additional use of the corpora, and add the missing relevant references.

---

> > ### Comment · Reviewer_iBch · 2022-08-08
> > **Thanks**
> >
> > Thanks for your clarification. I have no other questions.

---

### Official Review · Reviewer_pYgq · 2022-07-12

**Rating:** 7
**Confidence:** 3
**Soundness:** 4 excellent
**Presentation:** 4 excellent
**Contribution:** 3 good

**Summary:**

The paper proposes a latent adaptive structure-aware generative language model for information extraction tasks. The proposed model incorporates a latent structure induction module that automatically induces tree-like structures akin to dependency and constituency trees. Experiments in IE benchmarks show that the proposed model outperforms the state-of-the-art models.

**Questions:**

* How different/similar are the induced trees and the original dependency/constituency trees? These can be compared by calculating the tree induction accuracy.

**Limitations:**

* A more extensive analysis on the induced trees would have improved the paper.

**Strengths And Weaknesses:**

Strengths
* I appreciate the extensiveness of the experiments in Section 5. I personally had questions regarding the difference between constituency and dependency structures as well as the difference between internally learned (latent) and externally predicted structures. These are answered well in the paper.
* The paper is written well. The motivation behind the use of latent structures and the intuition behind the design of the model look sound.

Weaknesses
* The idea of using task-specific latent structures started in this paper [1], however this is not mentioned at all in the paper. Moreover, there are many prior work in latent structure induction [2, 3, among others] that were also not mentioned in the paper.
* It would be very helpful for readers to know the tradeoff between the performance and speed. This is crucial especially since the differences in performance between the proposed model and the state-of-the-art models are rather small.
* Please also include significance testing in the results.
* A more extensive analysis on the induced tree structures would also be very useful for readers. For example, are the trees interpretable and useful outside of the model? Can humans make use of such trees to explain predictions? Also, including examples of induced trees would be nice to have.

[1] https://ojs.aaai.org/index.php/AAAI/article/view/11975
[2] https://arxiv.org/abs/1904.02142
[3] https://direct.mit.edu/tacl/article/doi/10.1162/tacl_a_00019/43445/Do-latent-tree-learning-models-identify-meaningful

---

> ### Author Response · Authors · 2022-07-31
> **Response to Reviewer pYgq**
>
> Thank you for acknowledging the strengths of our work. Following we show the feedbacks on your concerns and questions.
>
> ----
>
> **Q: Prior related work of task-specific latent structure idea is not mentioned in the paper.**
>
> **A:** Thank you for indicating these prior works. In appendix **A.4 Extended Related Work** we mentioned the line of work about the latent structure induction [55,56,23]. We will further add the references in our revision about the idea of constructing task-specific latent structures you pointed out here [1,2,3], and clearly state them in the Related Work part.
>
> [1] Jihun Choi, Kang Min Yoo, Sang-goo Lee. Learning to Compose Task-Specific Tree Structures. In AAAI 2018: 5094-5101.
>
> [2] Andrew Drozdov, Patrick Verga, Mohit Yadav, Mohit Iyyer, Andrew McCallum. Unsupervised Latent Tree Induction with Deep Inside-Outside Recursive Auto-Encoders. In NAACL-HLT (1) 2019: 1129-1141.
>
> [3] Adina Williams, Andrew Drozdov, Samuel R. Bowman. Do latent tree learning models identify meaningful structure in sentences? TACL 6: 253-267 (2018).
>
> [23] Yoon Kim, Chris Dyer, and Alexander Rush. Compound probabilistic context-free grammars for grammar induction. In ACL, pages 2369–2385, 2019.
>
> [55] Yikang Shen, Zhouhan Lin, Chin-Wei Huang, and Aaron C. Courville. Neural language modeling by jointly learning syntax and lexicon. In ICLR, 2018.
>
> [56] Yikang Shen, Shawn Tan, Alessandro Sordoni, and Aaron C. Courville. Ordered neurons: Integrating tree structures into recurrent neural networks. In ICLR, 2019.
>
> ----
>
> **Q: The tradeoff between the performance and speed.**
>
> **A:** We in appendix **D.2 Efficiency Analysis** presented the discussion concerning the model inference speed, and made comparisons with the SoTA baselines. We show that our structure-aware GLM that makes use of the latent structures achieves better performances without the sacrifice of running efficiency much. We kindly refer the reviewer to this part for a more detailed analysis.
>
>
> ----
>
> **Q: Include significance testing.**
>
> **A:** We in appendix **C.3.4 Evaluation** detailed the specification of experimental evaluation. We report the average scores with unbiased standard deviations on 5 runs with different random seeds. In practice, for those re-implemented baselines we actually performed **paired t-test** with $p < 0.05$.
>
>
> ----
>
> **Q: A more extensive analysis of the induced tree structures would also be very useful.**
>
> **A:** We in appendix **D.6 Case Study** presented some pieces of empirical visualizations of the induced structures. From the visualizations we also find that our LasUIE GLM has the advantage of explainable prediction.
>
>
>
> ----
>
> **Q: How different/similar are the induced trees and the original dependency/constituency trees?**
>
> **A:** To validate this, we in these rebuttal days perform the unsupervised tree (grammar) induction based on the PTB test set, including the Constituency tree and Dependency tree, and make comparisons (Accuracy) with some representative methods of this task. Note that we directly take the induced trees from the attention heads before compacting them into the forest so that we retain the tree topology. We see from the table that our model after post-training of unsupervised structure induction shows the tree induction performances on par with strong-performing systems. After fine-tuning with the end tasks, interestingly, the results of tree induction are rapidly dropped. We assume that there are quite distinctions between the induced trees (structures) and the fixed dependency/constituency trees.
>
> |  Model | Constituency | Dependency |
> |  -  | :-:  | :-: |
> | PRPN[1]  | 58.3 | / |
> | PCFG[2]  | 60.1 | / |
> | DIORA[3]  | 56.2 | / |
> | NDMV[4]  | / | 67.5 |
> | LasUIE (after post-training)  | 53.6 | 64.4 |
> | LasUIE (after fine-tuning)  | 17.2 | 25.6 |
>
>
> [1] Yikang Shen, Zhouhan Lin, Chin-Wei Huang, and Aaron C. Courville. Neural language modeling by jointly learning syntax and lexicon. In ICLR, 2018.
>
> [2] Yoon Kim, Chris Dyer, and Alexander Rush. Compound probabilistic context-free grammars for grammar induction. In ACL, pages 2369–2385, 2019.
>
> [3] Andrew Drozdov, Patrick Verga, Mohit Yadav, Mohit Iyyer, Andrew McCallum. Unsupervised Latent Tree Induction with Deep Inside-Outside Recursive Auto-Encoders. In NAACL-HLT (1) 2019: 1129-1141.
>
> [4] Songlin Yang, Yong Jiang, Wenjuan Han, Kewei Tu. Second-Order Unsupervised Neural Dependency Parsing. In COLING 2020: 3911-3924.

---

> > ### Comment · Reviewer_pYgq · 2022-08-08
> > **Thanks for the response!**
> >
> > Thanks. I think all your responses make sense. For the last question, I think it would be more informative if you could present such numbers per constituency/dependency tags.

---

> > > ### Author Response · Authors · 2022-08-09
> > > **Re-response**
> > >
> > > Thank you again for your acknowledgment. We representatively present the parsing results of the constituency syntax. Following are the experimental results of the grammar induction w.r.t. each tag, as you indicated. The results are the recall rates of the labels that were identified by the model (label recall).
> > >
> > >
> > > | Model | SBAR | NP    | VP   | PP    | ADJP | ADVP |
> > > |  ---- | ---- | ----  | ---- | ----  | ---- | ---- |
> > > | PRPN  | 50.0 | 59.2  | 46.7 | 57.2  | 44.3 | 32.8 |
> > > | PCFG  | 56.1 | 74.7  | 41.7 | 48.8  | 40.4 | 52.5 |
> > > | DIORA | 53.2 | 65.7  | 45.2 | 59.0  | 40.1 | 29.7 |
> > > | LasUIE (after post-training)	  | 47.3 | 53.7  | 37.8 | 43.8  | 36.4 | 27.0 |
> > > | LasUIE (after fine-tuning)	  |  7.5 | 30.2  | 16.5 | 15.6  | 4.7  | 8.2 |
> > >
> > > We see that the fine-tuned LM shows clearly better parsing results on those shorter phrases (e.g., NP, VP, PP), instead of the long expressions (e.g., SBAR, ADJP, ADVP). We can imagine the fine-tuned model learns fine-grained phrases that are coincident with the end tasks' needs.

---

### Author Response · Authors · 2022-07-31
**Response to All Reviewers**

We thank all reviewers for their time and for giving valuable and supportive comments. Information Extraction is one of the most fundamental tasks in NLP & data mining community. In this paper we follow the latest line of IE under the unified form (i.e., UIE), and investigate a generative language model that is empowered with linguistic(-like) structure knowledge. Both theoretically and empirically, our method shows great potential in solving the key challenges of the IE or UIE, including the long-range dependence issue and boundary identifying, pushing the current state of the art on a wide range of IE datasets.

We believe our method will spur more follow-up research on the line of structure-aware LM for UIE, and meanwhile show more impacts on NLP community. We will release our codes and metadata upon the acceptance of the work. We will further proofread the article, correct all the typos and double-check the contents according to reviewers’ comments, so as to make it ready to publish.

Additionally, we would like to draw reviewers’ attention to the [**supplementary material**](https://openreview.net/attachment?id=a8qX5RG36jd&name=supplementary_material) part. There we previously uploaded the full version of our paper with **detailed appendix**, including many more model and experimental specifications and extended analyses. We sincerely hope that reviewers could have a read, which may help to build a better understanding of this work.

---

### Meta-Review · Area_Chair_PJ8g · 2022-08-26

**Recommendation:** Accept
**Confidence:** Certain

**Metareview:**

This paper proposes a latent adaptive structure-aware generative language model (GLM) to leverage syntactic knowledge for information extraction tasks. The proposed model incorporates a latent structure induction module that automatically induces tree-like structures akin to dependency and constituency trees. Experiments in 12 IE benchmarks across 7 tasks showed significant improvements over the baseline.

Overall, all reviewers feel positively about this paper, even though they mention some aspects which can be improved in the final version. The conversion of information extraction tasks into a problem solvable by a GLM problem with three different prediction modules is original and valuable, and the experiments are well designed and generally convincing (although additional experiments in more recent and larger scale datasets would make the paper stronger). The author response addressed well all the concerns of the reviewers, including the addition of several missing references. I urge the authors to incorporate these in their paper and to report the runtime of their method, to better understand the tradeoff between performance and speed, as well as examples of induced tree structures produced by their method, as suggested by one of the reviewers.

**Award:**

No

---

### Decision · Program_Chairs · 2022-09-14

Accept